# Tombstone cost and longevity: The San Pedro Cemetery Museum in Medellín in Colombia

Gary O'Donovan[1,2,3]*, Astrid Martínez[4], Santiago Flórez[5], Ana Isabel Cadavid Castrillón[6], Uriel Zapata[4]

1 Facultad de Medicina, Universidad de los Andes, Bogotá, Colombia, 2 Latin American Brain Health Institute (BrainLat), Universidad Adolfo Ibáñez, Santiago, Chile, 3 Instituto Masira, Facultad de Ciencias Médicas, Universidad de Santander (UDES), Bucaramanga, Colombia, 4 School of Applied Sciences and Engineering, EAFIT University, Medellín, Colombia, 5 Fundación Universitaria Bellas Artes, Medellín, Colombia, 6 Independent Researcher, Medellín, Colombia

* g.odonovan@uniandes.edu.co

## Abstract

### Background

Studies in the West suggest that tombstone cost is associated with longevity. The objective of this observational study was to investigate the association between tombstone cost and longevity in a large cemetery in Latin America.

### Methods

Age at death was obtained from 2,273 consecutive death certificates held at the San Pedro Cemetery Museum in Medellín in Colombia. Subjects died in 2022, 2021, or 2020. Tombs are arranged in galleries in the cemetery and tombstone cost was based on the material from which the tombstone was made, its position in the gallery, and its ornamentation. Analysis of variance was used and the assumption of equal variance was not violated.

### Results

Approximately 77% of tombstones were of low cost, 21% of medium cost, and 2% of high cost. Data from 1,751 subjects were used to investigate differences in longevity according to tombstone cost while adjusting for sex, civil status, violent death, and year of death. Longevity was similar in the low-cost group and medium-cost group: 64.3 years (63.2, 65.3) versus 63.3 years (61.3, 65.3) [estimated mean (95% confidence interval)]. Longevity was lower in the high-cost group: 47.0 years (40.1, 53.9).

### Conclusions

The inverse association between tombstone cost and longevity would suggest that people in Medellín are inclined to spend more on tombstones when commemorating the tragic death of a young person.

**Data Availability Statement:** The data underlying the results presented in the study can be found on

the public repository, UK Data Service ReShare:
https://reshare.ukdataservice.ac.uk/856516/.

**Funding:** The authors received no specific funding
for this work.

**Competing interests:** The authors have declared
that no competing interests exist.

## Introduction

The few studies that exist suggest that tombstone cost is associated with longevity [1,2]. Davey Smith and colleagues assumed that taller obelisks cost more than shorter obelisks and they investigated the association between obelisk height and longevity in eight cemeteries in Glasgow in Scotland [1]. Data were obtained from obelisks built from 1801 to 1920 and it was found that obelisk height was correlated with age at death [1]. Kang-Auger and colleagues assumed that larger tombstones cost more than smaller tombstones and they investigated the association between tombstone size and longevity in a cemetery in Quebec in Canada [2]. Data were obtained from tombstones built from 1820 to 1992 and it was found that tombstone height and tombstone volume were associated with age at death after adjustment for sex, marital status, and year of death [2]. There was a lack of good quality data in Scotland and Canada in the nineteenth and early twentieth centuries and these studies were inspired by the need to find alternative ways of investigating associations between socioeconomic status and longevity [1,2]. There is still relatively little known about associations between socioeconomic status and longevity in Latin America because of the lack of good quality data in the region [3]. The objective of the present study was to investigate the association between tombstone cost and longevity in a large cemetery in Medellín in Colombia. We hypothesized that tombstone cost would be positively associated with longevity.

## Materials and methods

### Setting and subjects

This study was conducted in the San Pedro Cemetery Museum in the city of Medellín in Colombia. The cemetery was founded in 1842 and was recognized as a museum in 1998 and as a national heritage site in 1999 [4]. A body may remain interred in the cemetery in perpetuity if the family owns the tomb. However, most tombs are rented. The cemetery has more than 10,000 tombs for rent and the Ministry of Health stipulates that a body must remain interred for at least three years if the deceased was less than seven years old and for at least four years if the deceased was more than seven years old. After the minimum period of interment, the family can choose to transfer the body to another tomb or to cremate the body and store the ashes in a columbarium. Many families take out an insurance plan to cover the various costs associated with the death of a loved one. Subjects in the present study were people who died in 2022, 2021, or 2020. Forty-seven deaths from COVID-19 were not included in the present study because the bodies had not been returned to the next of kin. Medellín had a Multidimensional Poverty Index of 13.44 in 2021 [5]. Medellín was also one of the riskiest cities in the world for murder and other crimes in 2022 [6].

### Variables

Data were obtained from 2,273 consecutive death certificates, starting with the most recently available. The dependent variable was longevity, defined as age at death. The independent variable was tombstone cost. Tombs are arranged in galleries in the San Pedro Cemetery Museum, as shown in Fig 1. There is no single measure of tombstone cost, so we created a scoring scheme based on the material from which the tombstone was made, the position of the tomb in the gallery, and the ornamentation added to the tombstone. For example, quartz tombstones, granite tombstones, and marble tombstones were scored higher than laminated tombstones. Tombstones at head height were scored higher than tombstones above or below head height. Tombstones with engravings and embellishments were scored higher than plain tombstones. Tombstones were classified as being of low cost, medium cost, or high cost as described

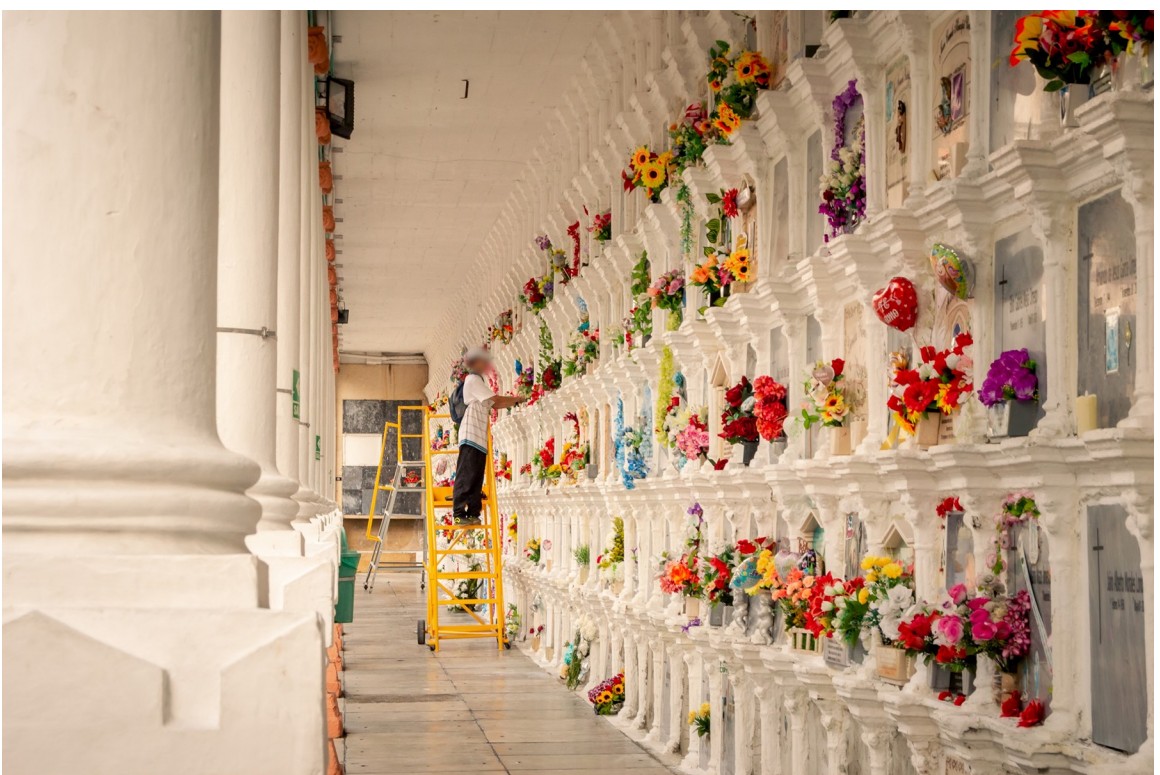

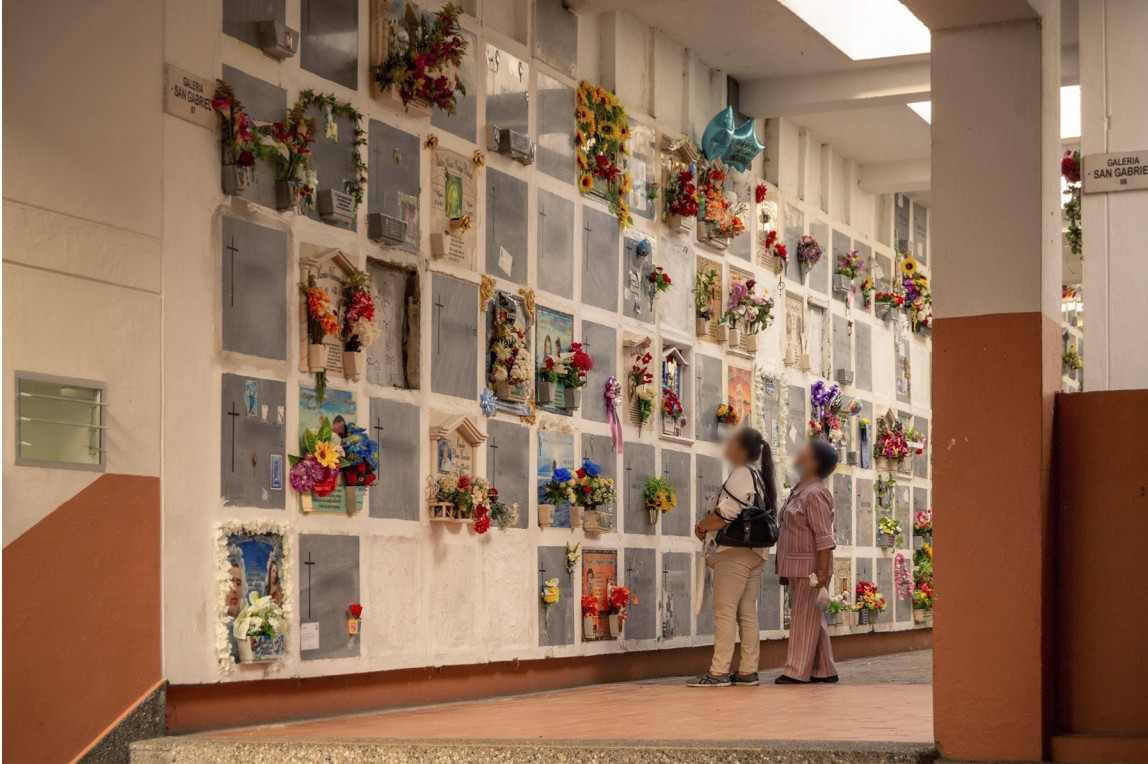

**Fig 1. Tombs are arranged in galleries in the San Pedro Cemetery Museum in Medellín in Colombia.** The figure shows Gallery 1, Los Dolores, and Gallery 87, San Gabriel (top and bottom, respectively). Photographs by Santiago Flórez (@santi.ph6).

in detail in the online supplement. Each death certificate had a code that indicated the location of the tomb in the cemetery. We created a mobile phone app to help us match up death certificates and tombstones, as described in detail in the online supplement.

In observational studies, a confounding factor may be defined as a prognostic factor that has the potential to introduce material bias into the relationship between the exposure and the outcome [7]. Material bias is considered as bias sufficient to affect the direction of the estimated effect or to impact on the ability to draw valid conclusions about the relationship between the exposure (tombstone cost) and the outcome (longevity) [7]. Confounding factors that may introduce material bias into the relationship between tombstone cost and longevity include sex, civil status, violent death, and year of death [2,3,8]. Information about potential confounders was taken from the death certificates.

## Statistical analysis

All analyses were performed using Stata SE version 17.0 for Mac (StataCorp, Texas, USA). We were not aware of any studies of tombstone cost and longevity in Latin America. Therefore, we had to use data from elsewhere to estimate sample size. We used data from Kang-Auger and colleagues' study of 276 people buried between 1820 and 1992 in a cemetery in Quebec in Canada [2]. We assumed that the standard deviation in longevity was 20 years [2]. We also assumed that mean life expectancy was 60 years in those buried in tombs of low cost, 63 years in those buried in tombs of medium cost, and 70 years in those buried in tombs of high cost [2]. These assumptions are broadly in keeping with life expectancy in Colombia during the period of investigation [9]. For example, life expectancy at birth in Colombia was around 60 years in 1960 [9]. We used the *twomeans* command in Stata to estimate sample size, where power was 0.8 and alpha was 0.05. The estimated sample size was 699 tombstones of low cost, 699 of medium cost, and 130 of high cost. Analysis of variance was used in the main analyses because the dependent variable was continuous and the independent variable was categorical. First, analysis of variance was used to investigate differences in longevity according to tombstone cost. Longevity was a continuous variable (age at death) and tombstone cost was a categorical variable (low versus medium versus high). The Bonferroni correction was used to adjust for multiple comparisons. Bartlett's test for equal variance was not statistically significant (p = 0.402), which suggests that the assumption of equal variance was not violated. Second, analysis of covariance was used to investigate differences in longevity according to tombstone cost while adjusting for potential confounders: sex (male versus female), civil status (not married or with partner versus married or with partner), violent death (not violent versus violent or under investigation), and year of death (continuous).

## Results

Table 1 shows subject characteristics. The mean age of death was 61 years. Approximately 77% of tombstones were of low cost, 21% of medium cost, and 2% of high cost. Approximately 56% of subjects were male. Around one third of deaths were violent or under investigation. There were few missing values other than civil status, which was not reported in 514 cases.

Table 2 shows the analysis of variance summary table. Data from 2,244 of 2,273 (99%) subjects were used to investigate differences in longevity according to tombstone cost. The analysis of variance was statistically significant: F(2,2241) = 10.56, p<0.001. Longevity was 61.6 ±23.9 years in the low-cost group and 61.1±25.0 years in the medium-cost group (mean±standard deviation) (p = 1.0). Longevity was 45.0±22.9 years in the high-cost group (p<0.001 versus low-cost group and medium-cost group).

**Table 1. Subject characteristics*.**

| Characteristic | | Value |
|---|---|---|
| Age at death, mean±SD (n) | | 61.1±24.2 (2,244) |
| Tombstone cost | | |
| | Low, n (%) | 1,751 (77.03) |
| | Medium, n (%) | 474 (20.85) |
| | High, n (%) | 48 (2.11) |
| | Missing, n (%) | 0 (0) |
| | Total, n (%) | 2,273 (100) |
| Sex | | |
| | Male, n (%) | 1,274 (56.05) |
| | Female, n (%) | 999 (43.95) |
| | Missing, n (%) | 0 (0) |
| | Total, n (%) | 2,273 (100) |
| Civil status | | |
| | Not married or with partner, n (%) | 1,092 (48.04) |
| | Married or with partner, n (%) | 667 (29.34) |
| | Missing, n (%) | 514 (22.61) |
| | Total, n (%) | 2,273 (100) |
| Violent death | | |
| | Not violent, n (%) | 1,531 (67.36) |
| | Violent or under investigation, n (%) | 725 (31.90) |
| | Missing, n (%) | 17 (0.75) |
| | Total, n (%) | 2,273 (100) |
| Characteristic | | Value |
| Year of death | | |
| | 2022, n (%) | 480 (21.12) |
| | 2021, n (%) | 1,154 (50.77) |
| | 2020, n (%) | 639 (28.11) |
| | Missing, n (%) | 0 (0) |
| | Total, n (%) | 2,273 (100) |

*Values are from 2,273 consecutive death certificates, starting with the most recently available. SD is standard deviation. N is number.

Table 3 shows the analysis of covariance summary table. Data from 1,751 subjects were used to investigate differences in longevity according to tombstone cost while adjusting for sex, civil status, violent death, and year of death. Longevity was similar in the low-cost group and medium-cost group: 64.3 years (63.2, 65.3) versus 63.3 years (61.3, 65.3) [estimated mean (95% confidence interval)]. Longevity was lower in the high-cost group: 47.0 years (40.1, 53.9).

**Table 2. Analysis of variance summary table*.**

| Source | SS | df | MS | F | Prob > F |
|---|---|---|---|---|---|
| Between groups | 12270.8465 | 2 | 6135.42323 | 10.56 | <0.001 |
| Within groups | 1301779.17 | 2241 | 580.892087 | | |
| Total | 1314050.01 | 2243 | 585.844856 | | |

*Analysis of variance was used to investigate differences in longevity (continuous variable) according to tombstone cost (categorical variable). SS is the sum of squared deviations for each source, df is degrees of freedom, MS is mean squares, and F is the test statistic.

**Table 3. Analysis of covariance summary table\*.**

| Source | Partial SS | df | MS | F | Prob > F |
|---|---|---|---|---|---|
| Model | 290944.96 | 6 | 48490.826 | 126.96 | <0.001 |
| Tombstone cost | 2639.0253 | 2 | 1319.5127 | 3.45 | 0.032 |
| Sex | 10567.53 | 1 | 10567.53 | 27.67 | <0.001 |
| Civil status | 3016.3144 | 1 | 3016.3144 | 7.90 | 0.005 |
| Violent death | 186538.07 | 1 | 186538.07 | 488.39 | <0.001 |
| Year of death | 28.779018 | 1 | 28.77901 | 0.08 | 0.784 |
| Residual | 666110.51 | 1,744 | 381.9441 | | |
| Total | 957055.47 | 1,750 | 546.88884 | | |

\*Analysis of covariance was used to investigate differences in longevity (continuous variable) according to tombstone cost (categorical variable), while adjusting for sex (categorical variable), civil status (categorical variable), violent death (categorical variable), and year of death (continuous variable). SS is the sum of squared deviations for each source, df is degrees of freedom, MS is mean squares, and F is the test statistic.

## Discussion

The objective of this study was to investigate the association between tombstone cost and longevity in a large cemetery in Medellín in Colombia. The results did not support the hypothesis that tombstone cost was positively associated with longevity. On the contrary, the most expensive tombstones were those of the young. The inverse association between tombstone cost and longevity would suggest that people in Medellín are inclined to spend more on tombstones when commemorating the tragic death of a young person.

Studies in the West support the hypothesis that tombstone cost is positively associated with longevity [1,2]. The height of 843 obelisks from the nineteenth and early twentieth centuries was associated with longevity in a study of eight cemeteries in Glasgow in Scotland [1]. The size of 165 tombstones from the nineteenth and twentieth centuries was also associated with longevity in a study of one large cemetery in Quebec in Canada [2]. This study in Latin America supports the alternative hypothesis that tombstone cost is inversely associated with longevity. We investigated 2,273 tombstones from 2020 to 2022 in the San Pedro Cemetery Museum in Medellín in Colombia. We found that the most expensive tombstones were those of the young. Socioeconomic status was not assessed in the study in Glasgow or the study in Quebec, but it was assumed that affluent families spent more to commemorate the deaths of wealthy individuals [1,2]. Socioeconomic status was not assessed in the present study either and we can only assume that ordinary families spend more to commemorate the deaths of young individuals.

It is quite normal to find tombstones of the young in the San Pedro Cemetery Museum in Medellín. Many of the young will have lost their lives in violent deaths, including gun-related deaths, knife-related deaths, traffic accidents, and suicides. The threat of death is so great in the poorest and most marginalized neighbourhoods that it is said colloquially that they are born with a tombstone on their back (*Nacen con la lápida pegada*, in Spanish). Drug dealing and violence have been commonplace in Medellín since the 1980's [10]. Many young people in marginalized communities do not have father figures and they think of drug dealing and violence as they only ways of becoming successful [10]. Gangs run many neighbourhoods and there is an eye-for-an-eye philosophy that results in an endless cycle of violence [10]. Violence and violent deaths are also part of the folklore of many communities in Medellín [10,11]. While many of the young subjects in the present study will not have been gang members, the glorification of violence may help explain why the most expensive tombstones were those of the young.

Medellín changed from being a small town to a city in the 1970's [12]. The population of Medellín doubled between 1965 and 1985 because of three waves of migrants: families fleeing political violence in the countryside; people looking to work and study and people looking for a better quality of life; and, people attracted by the government's promotion of urban developments [12]. The authorities struggled to cope with the growing population and there was an industrial crisis, high unemployment, and a large black market controlled by drug-traffickers [12]. Medellín was characterized by two phenomena of urban growth in the 1970's: there was planned and regulated urbanization along the west side of the River Medellín and there was informal and illegal development on the hillsides of the north of the city [12]. More than 57,000 young people were murdered in Medellín between 1979 and 2018 and people aged 18 to 24 years are still the main victims of murder [13]. This long cycle of violence has led many academics to conclude that the poor and the marginalized are destined to join a gang and die young [14]. Murder is the ultimate manifestation of taking the law into one's own hands in poor and marginalized communities in Medellín and elsewhere in Latin America [14] and an investigation of 12 young murder victims found that 11 of them had experienced threats and violent situations weeks, months, and even years before their deaths [15]. It is the precarious nature of many young people's lives in Medellín that leads to violence [14]. Precarious economic conditions mean that many young people have limited access to essential goods and services [14]. Precarious social conditions mean that many young people do not enjoy the same social fabric and the same social mobility as others [14]. Precarious cultural conditions mean that many young people lack appropriate role models and are exposed to perverse norms and values [14]. Precarious access to formal justice means that violence is often used to settle disputes [14]. More than 200 young people in Medellín took part in workshops and semi-structured interviews between 2018 and 2019, including young people who had committed murder [14]. Nine external factors were found to be associated with violent crime and murder: selling and consuming narcotic drugs; the need or desire to earn money; dropping out of school; displacement by violence; coming from a violent family or violent community; living in a community run by gangs; 'invisible borders' and limited mobility between communities; the opportunity to join a local gang; and, the exposure to uncontrolled violence [14]. Four profiles of young people at risk of being the victim or the perpetrator of murder were also identified: members of criminal gangs; members of less formal groups of drug users and petty criminals; independent criminals who are not affiliated with a gang; and, young people outside of the criminal world who kill after taking drugs or getting into an argument [14]. Gang-related crime was the main cause of murder in young people in Medellín in 2020, 2021, and 2022 [13], but we have no way of knowing whether the young subjects in the present study were victims of gang-related crime.

This study has strengths and limitations. To the best of our knowledge, this is the largest study of the association between tombstone cost and longevity and the only such study in Latin America. Age at death was obtained from death certificates, tombstone cost was carefully estimated, and the analyses were adjusted for potential confounders. The number of subjects in the high-cost group was lower than planned, but the relatively narrow confidence intervals in each group would suggest that the study was large enough to give precise results. Civil status was missing from 514 subjects, which could cause bias if the missing subjects were children or young adults. However, the results were not changed when missing values were coded as not married or with partner [age at death was 61.8 years (60.9, 62.7) in the low-cost group, 61.1 years (59.4, 62.9) in the medium-cost group, and 45.6 years (40.0, 51.3) in the high-cost group]. Medellín may have been more affected by drug dealing and violence than any other city in the last 50 years [10,16,17]; And, the present study offers a unique snapshot of life and death in Medellín in recent years. Consistency is an important consideration in epidemiology

[18] and more research should be conducted in other cemeteries to investigate whether tombstone cost is positively or negatively associated with longevity.

In conclusion, the present study does not support the hypothesis that tombstone cost is positively associated with longevity. Rather, the present study supports the alternative hypothesis that tombstone cost is inversely associated with longevity. This is the largest study of its kind and the results would suggest that people in the city of Medellín in Colombia are inclined to spend more on tombstones when commemorating the tragic death of a young person.

## Supporting information

**S1 Checklist. STROBE statement—checklist of items that should be included in reports of observational studies.**
(DOCX)

**S1 File. Online supplement, tombstone cost and longevity.**
(PDF)

## Acknowledgments

We thank Valentina Móvil Sandoval for her help in developing the mobile phone app used to facilitate data collection. We also thank Luciana Zapata for her help with data collation.

## Author Contributions

**Conceptualization:** Gary O'Donovan, Ana Isabel Cadavid Castrillón.

**Data curation:** Gary O'Donovan, Astrid Martínez, Santiago Flórez, Uriel Zapata.

**Formal analysis:** Gary O'Donovan.

**Investigation:** Gary O'Donovan, Astrid Martínez, Santiago Flórez, Ana Isabel Cadavid Castrillón, Uriel Zapata.

**Methodology:** Gary O'Donovan, Astrid Martínez, Uriel Zapata.

**Project administration:** Gary O'Donovan.

**Supervision:** Gary O'Donovan, Uriel Zapata.

**Writing – original draft:** Gary O'Donovan, Astrid Martínez.

**Writing – review & editing:** Gary O'Donovan, Uriel Zapata.

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
