## [Editor Report · Decision Letter 0]

27 Apr 2023

PONE-D-23-07060Tombstone cost and longevity: the San Pedro Cemetery Museum in Medellín in ColombiaPLOS ONE

Dear Dr. O'Donovan,

Thank you for submitting your manuscript to PLOS ONE. After careful consideration, we feel that it has merit but does not fully meet PLOS ONE’s publication criteria as it currently stands. Therefore, we invite you to submit a revised version of the manuscript that addresses the points raised during the review process.

 Please included in the manuscript the details of the ANCOVA model, the reason to use that approach, and the results in detail (table).

We look forward to receiving your revised manuscript.

Kind regards,

Juan Pablo Gutierrez

Academic Editor

PLOS ONE

Journal Requirements:

Additional Editor Comments (if provided):

Please include the results from your analysis controlling for sex, civil status, violent death, and year of death (mentioned in the manuscript but not reported in tables). In the methods, please proved further detail on the ANCOVA model and why you opted for that model.
---

## [Author Response · Author response to Decision Letter 0]

24 May 2023

Tombstone cost and longevity: the San Pedro Cemetery Museum in Medellín in Colombia

Response to reviewers

Reviewer comments

[Comment 1] Please included in the manuscript the details of the ANCOVA model, the reason to use that approach, and the results in detail (table).

[Response 1] In the revised manuscript, it now says: “Analysis of variance was used in the main analyses because the dependent variable was continuous and the independent variable was categorical.” We have also added two tables. Table 2 is the analysis of variance summary table. Table 3 is the analysis of covariance summary table.

Editor comments

[Comment 1] Please include the results from your analysis controlling for sex, civil status, violent death, and year of death (mentioned in the manuscript but not reported in tables). In the methods, please proved further detail on the ANCOVA model and why you opted for that model.

[Response 1]. In the revised manuscript, we explain why we used analysis of variance: “Analysis of variance was used in the main analyses because the dependent variable was continuous and the independent variable was categorical.” We have also added detailed results of the analyses; please see Table 2 and Table 3.

---

## [Decision Letter · Decision Letter 1]

13 Jun 2023

PONE-D-23-07060R1Tombstone cost and longevity: the San Pedro Cemetery Museum in Medellín in ColombiaPLOS ONE

Dear Dr. O'Donovan,

Thank you for submitting your manuscript to PLOS ONE. After careful consideration, we feel that it has merit but does not fully meet PLOS ONE’s publication criteria as it currently stands. Therefore, we invite you to submit a revised version of the manuscript that addresses the points raised during the review process. As mentioned in the previous iteration, it would be beneficial to provide further details on the methodological approach, in particular, its limitations and how those limitations may affect your results. In addition, your discussion should take into account the limitations of the study. It seems that relevant aspects (see reviewers' comments below) that could affect your results were not fully considered, such as the relationship between gangs and tombstone costs.

We look forward to receiving your revised manuscript.

Kind regards,

Juan Pablo Gutierrez

Academic Editor

PLOS ONE

Reviewers' comments:

Reviewer's Responses to Questions

**Comments to the Author**

1. If the authors have adequately addressed your comments raised in a previous round of review and you feel that this manuscript is now acceptable for publication, you may indicate that here to bypass the “Comments to the Author” section, enter your conflict of interest statement in the “Confidential to Editor” section, and submit your "Accept" recommendation.

Reviewer #1: All comments have been addressed

Reviewer #2: (No Response)

2. Is the manuscript technically sound, and do the data support the conclusions?

Reviewer #1: Yes

Reviewer #2: Partly

3. Has the statistical analysis been performed appropriately and rigorously? 

Reviewer #1: Yes

Reviewer #2: Yes

4. Have the authors made all data underlying the findings in their manuscript fully available?

Reviewer #1: Yes

Reviewer #2: Yes

5. Is the manuscript presented in an intelligible fashion and written in standard English?

Reviewer #1: Yes

Reviewer #2: Yes

6. Review Comments to the Author

Reviewer #1: (No Response)

Reviewer #2: Dear Author(s),

I have gone through your manuscript and would like to raise a few points of clarification:

1. Your hypothesis revolves around the relationship between age and tombstone cost. It might add depth to your study if you could perform the analysis segmented by different age groups. Perhaps you could consider delineating groups such as children, young adults, adults, and the elderly, and examine how the cost varies across these groups.

2. It appears that you intended to estimate a certain sample size but the actual number of samples you obtained does not match this. Could you please clarify why you were unable to acquire the estimated number of samples?

3. Your analysis controls for gender, but the sample size seems particularly small for high-cost male tombstones, with only 9 examples, and only one of these is under 36 years old. Could you comment on how this limited sample size might influence the robustness of your findings?

4. You also control for civil status in your study, yet this control variable seems inappropriate for some categories of your data, notably when considering children. How did you address this issue within your analysis?

5. Finally, your claim that "the glorification of violence may help explain why the most expensive tombstones were those of the young" raises a question. How do you reconcile this statement with your methodology, specifically when you are controlling for violent deaths? How does this control affect your conclusions?

Thank you for considering these questions. I believe your responses will greatly enhance the clarity and robustness of your research.

7. PLOS authors have the option to publish the peer review history of their article (what does this mean?). If published, this will include your full peer review and any attached files.

Reviewer #1: No

Reviewer #2: No

---

## [Author Response · Author response to Decision Letter 1]

26 Jun 2023

Tombstone cost and longevity: the San Pedro Cemetery Museum in Medellín in Colombia

Response to reviewers

Reviewer comments

[Comment 1] I have gone through your manuscript and would like to raise a few points of clarification. Your hypothesis revolves around the relationship between age and tombstone cost. It might add depth to your study if you could perform the analysis segmented by different age groups. Perhaps you could consider delineating groups such as children, young adults, adults, and the elderly, and examine how the cost varies across these groups.

[Response 1]. Thank you for considering our manuscript. We appreciate your constructive and insightful comments. Our hypothesis is about the association between tombstone cost (exposure) and age at death (outcome). We cannot perform the stratification the reviewer suggests because, in stratification, one can only separate the sample into sub-samples according to exposures or confounders, not outcomes (Porta, M. A Dictionary of Epidemiology. Sixth Edition. Oxford: Oxford University Press; 2014). 

It is also important to stress that the study was not designed to be divided into sub-samples, or strata. When a study is divided into strata, the efficiency (or precision) of the study can be affected dramatically (Rothman, K. J. and Lash, T. L. Precision and Study Size. In: Lash, T. et al., Modern Epidemiology. Fourth Edition Philadelphia: Wolters Kluwer; 2021). In other words, out study would have to be much bigger if it were to be divided into strata.

[Comment 2] It appears that you intended to estimate a certain sample size but the actual number of samples you obtained does not match this. Could you please clarify why you were unable to acquire the estimated number of samples?

[Response 2] We did indeed perform a sample size calculation to give us a rough idea of how many tombstones we might have to include. The estimated sample size was 1,528, including 699 tombstones of low cost, 699 of medium cost, and 130 of high cost. We actually created a sample of 2,273 tombstones before analyzing the data. We spent more than a year collecting the data and would have continued if the results had not been so clear. 

[Comment 3] Your analysis controls for gender, but the sample size seems particularly small for high-cost male tombstones, with only 9 examples, and only one of these is under 36 years old. Could you comment on how this limited sample size might influence the robustness of your findings?

[Response 3] We would like to take this opportunity to clarify our approach to designing the study and analyzing the data. In the Methods section, we have added a paragraph explaining that the analyses were adjusted for potential confounders that might impact on our ability to draw valid conclusions. Please notice that potential confounders were chosen with reference to the literature. In the revised manuscript, it now says: “In observational studies, a confounding factor may be defined as a prognostic factor that has the potential to introduce material bias into the relationship between the exposure and the outcome [ROBINS-E Development Group. Risk Of Bias in Non-randomized Studies of Exposure (ROBINS – E). 2022. Available online: https://www.riskofbias.info/welcome/robins-e-tool]. Material bias is considered as bias sufficient to affect the direction of the estimated effect or to impact on the ability to draw valid conclusions about the relationship between the exposure (tombstone cost) and the outcome (longevity) [ROBINS -E Development Group, 2022]. Confounding factors that may introduce material bias into the relationship between tombstone cost and longevity include sex, civil status, violent death, and year of death [Kang-Auger et al., Eur J Epidemiol. 2021, 36, 1219-23; Hambleton et al., PLoS One, 2015, 10, e129778; Aburto et al., Sci Adv. 2023, 9, eadd9038]. Information about potential confounders was taken from the death certificates.” In the revised manuscript, we now use phrases like ‘adjusted for’ rather than phrases like ‘controlling for’.

[Comment 4] You also control for civil status in your study, yet this control variable seems inappropriate for some categories of your data, notably when considering children. How did you address this issue within your analysis?

[Response 4] This is a very interesting comment. We agree that the missing data could cause bias and we have added a sensitivity analysis: “Civil status was missing from 514 subjects, which could cause bias if the missing subjects were children or young adults. However, the results were not changed when missing values were coded as not married or with partner [age at death was 61.8 years (60.9, 62.7) in the low-cost group, 61.1 years (59.4, 62.9) in the medium-cost group, and 45.6 years (40.0, 51.3) in the high-cost group].

[Comment 5] Finally, your claim that "the glorification of violence may help explain why the most expensive tombstones were those of the young" raises a question. How do you reconcile this statement with your methodology, specifically when you are controlling for violent deaths? How does this control affect your conclusions?

[Response 5] We hope that, in the revised manuscript, it is now clear that we wished to adjust for potential confounders, rather than control for them per se. We find it reassuring that the results of the ANOVA and ANCOVA analyses are similar because it suggests that the potential confounders did not have a great impact on the association between tombstone cost and longevity.

---

## [Editor Report · Decision Letter 2]

4 Sep 2023

PONE-D-23-07060R2

Tombstone cost and longevity: the San Pedro Cemetery Museum in Medellín in Colombia

PLOS ONE

Dear Dr. O'Donovan,

Thank you for submitting your manuscript to PLOS ONE. After careful consideration, we feel that it has merit but does not fully meet PLOS ONE’s publication criteria as it currently stands. Therefore, we invite you to submit a revised version of the manuscript that addresses the points raised during the review process.

We look forward to receiving your revised manuscript.

Kind regards,

Juan Pablo Gutierrez

Academic Editor

PLOS ONE

Journal Requirements:

Additional Editor Comments:

Thank you for submitting your manuscript to PLOS ONE. As discussed earlier, you need to incorporate the answers to both reviewers from the initial assessment. I apologize for the delay in sending you back the paper; it was totally outside of my control. The comments from reviewer #1 were provided as a separate Word file and this should be attached to this decision letter (the file is called: Rev PLOS One.docx). If it is not present please contact the Editorial Office. Please submit a revised version of the manuscript that addresses the points raised during the review process.

---

## [Author Response · Author response to Decision Letter 2]

5 Sep 2023

Tombstone cost and longevity: the San Pedro Cemetery Museum in Medellín in Colombia

Response to reviewers

Reviewer comments

Reviewer 1

[Comment 1] It is an interesting and novel study with data from a Latin American city. However, the manuscript lacks very relevant information about Medellin, the illegal activities that are generated from there. Some authors have classified Colombia as a case of a society with different behaviors from what is observed in other places, which is related to the drug trafficking and an armed conflict of several decades. Understanding Medellin society requires understanding the illegality associated with drug trafficking, a fact that may be changing the study's findings.

[Response 1] Thank you for considering our study. We are glad that you found our work interesting and thought-provoking. All of the authors of the study have lived and worked in Medellín for many years and we agree with the reviewer: Medellín is a unique place! In the original manuscript, we did discuss gangs, drug dealing, and violence. We even mentioned that many people are destined to suffer a violent death: “Nacen con la lápida pegada”. We also cited the import work of the Colombian journalist, María Jimena Duzán (Duzán MJ. Sicarios. Crónicas que matan. Bogotá, Colombia: Penguin Random House; 2012). In the revised manuscript, we mention that the novelty of the study is one of its strengths and we cite the important work the reviewer mentioned: “Medellín may have been more affected by drug dealing and violence than any other city in the last 50 years [Duzán MJ. Sicarios. Crónicas que matan. Bogotá, Colombia: Penguin Random House; 2012; Dellasega M, Vorrath J. A Gangster’s Paradise? Transnational Organised Crime in the Covid-19 Pandemic 2020; Sanchez Parra T. What’s killing them: Violence beyond COVID-19 in Colombia. Crime, Media, Culture. 2020;17(1):11-6]; And, the present study offers a unique snapshot of life and death in Medellín in recent years.” We think that any further speculation about the inverse association between tombstone cost and longevity in Medellín is beyond the scope of this study.

[Comment 2] From the methodological point of view, why were the years 2020, 2021 and 2022 chosen? This must be very well supported, because it may be related to the findings. There is news and some academic publications suggesting that illicit activities did not stop during the pandemic. Injuries and deaths among young people are consequences in this city. For example, see: Dellasega, Maria, and Judith Vorrath. "A Gangster’s Paradise?." Transnational Organised Crime in the Covid-19 Pandemic. SWP Comment, December 66 (2020). Tamayo Gomez C. Organised Crime Governance in Times of Pandemic: The Impact of COVID-19 on Gangs and Drug Cartels in Colombia and Mexico. Bulletin of Latin American Research 2020;39:12–15.

[Response 2] In the methods section, we explain that most tombs are rented for no longer than three or four years: “The cemetery has more than 10,000 tombs for rent and the Ministry of Health stipulates that a body must remain interred for at least three years if the deceased was less than seven years old and for at least four years if the deceased was more than seven years old. After the minimum period of interment, the family can choose to transfer the body to another tomb or to cremate the body and store the ashes in a columbarium.” For this reason, we had to use the most recent death certificates. 

[Comment 3] Other publications that can contribute to the discussion are suggested:

Fernández-Niño J et al. The other social capital: a needed look at Latin America. Rev Panam Salud Publica 2014;35:291-2. Rubio M. Perverse social capital – some evidence from Colombia. J Econ Issues. 1997;31:805–16. Sanchez Parra T. What’s killing them: Violence beyond COVID-19 in Colombia. Crime Media Culture 2021, Vol. 17(1) 11–16.

[Response 3]. We now cite the recent work of Sanchez Parra.

Reviewer 2

[Comment 1] I have gone through your manuscript and would like to raise a few points of clarification. Your hypothesis revolves around the relationship between age and tombstone cost. It might add depth to your study if you could perform the analysis segmented by different age groups. Perhaps you could consider delineating groups such as children, young adults, adults, and the elderly, and examine how the cost varies across these groups.

[Response 1]. Thank you for considering our manuscript. We appreciate your constructive and insightful comments. Our hypothesis is about the association between tombstone cost (exposure) and age at death (outcome). We cannot perform the stratification the reviewer suggests because, in stratification, one can only separate the sample into sub-samples according to exposures or confounders, not outcomes (Porta, M. A Dictionary of Epidemiology. Sixth Edition. Oxford: Oxford University Press; 2014). 

It is also important to stress that the study was not designed to be divided into sub-samples, or strata. When a study is divided into strata, the efficiency (or precision) of the study can be affected dramatically (Rothman, K. J. and Lash, T. L. Precision and Study Size. In: Lash, T. et al., Modern Epidemiology. Fourth Edition Philadelphia: Wolters Kluwer; 2021). In other words, out study would have to be much bigger if it were to be divided into strata.

[Comment 2] It appears that you intended to estimate a certain sample size but the actual number of samples you obtained does not match this. Could you please clarify why you were unable to acquire the estimated number of samples?

[Response 2] We did indeed perform a sample size calculation to give us a rough idea of how many tombstones we might have to include. The estimated sample size was 1,528, including 699 tombstones of low cost, 699 of medium cost, and 130 of high cost. We actually created a sample of 2,273 tombstones before analyzing the data. We spent more than a year collecting the data and would have continued if the results had not been so clear. 

[Comment 3] Your analysis controls for gender, but the sample size seems particularly small for high-cost male tombstones, with only 9 examples, and only one of these is under 36 years old. Could you comment on how this limited sample size might influence the robustness of your findings?

[Response 3] We would like to take this opportunity to clarify our approach to designing the study and analyzing the data. In the Methods section, we have added a paragraph explaining that the analyses were adjusted for potential confounders that might impact on our ability to draw valid conclusions. Please notice that potential confounders were chosen with reference to the literature. In the revised manuscript, it now says: “In observational studies, a confounding factor may be defined as a prognostic factor that has the potential to introduce material bias into the relationship between the exposure and the outcome [ROBINS-E Development Group. Risk Of Bias in Non-randomized Studies of Exposure (ROBINS – E). 2022. Available online: https://www.riskofbias.info/welcome/robins-e-tool]. Material bias is considered as bias sufficient to affect the direction of the estimated effect or to impact on the ability to draw valid conclusions about the relationship between the exposure (tombstone cost) and the outcome (longevity) [ROBINS -E Development Group, 2022]. Confounding factors that may introduce material bias into the relationship between tombstone cost and longevity include sex, civil status, violent death, and year of death [Kang-Auger et al., Eur J Epidemiol. 2021, 36, 1219-23; Hambleton et al., PLoS One, 2015, 10, e129778; Aburto et al., Sci Adv. 2023, 9, eadd9038]. Information about potential confounders was taken from the death certificates.” In the revised manuscript, we now use phrases like ‘adjusted for’ rather than phrases like ‘controlling for’.

[Comment 4] You also control for civil status in your study, yet this control variable seems inappropriate for some categories of your data, notably when considering children. How did you address this issue within your analysis?

[Response 4] This is a very interesting comment. We agree that the missing data could cause bias and we have added a sensitivity analysis: “Civil status was missing from 514 subjects, which could cause bias if the missing subjects were children or young adults. However, the results were not changed when missing values were coded as not married or with partner [age at death was 61.8 years (60.9, 62.7) in the low-cost group, 61.1 years (59.4, 62.9) in the medium-cost group, and 45.6 years (40.0, 51.3) in the high-cost group].

[Comment 5] Finally, your claim that "the glorification of violence may help explain why the most expensive tombstones were those of the young" raises a question. How do you reconcile this statement with your methodology, specifically when you are controlling for violent deaths? How does this control affect your conclusions?

[Response 5] We hope that, in the revised manuscript, it is now clear that we wished to adjust for potential confounders, rather than control for them per se. We find it reassuring that the results of the ANOVA and ANCOVA analyses are similar because it suggests that the potential confounders did not have a great impact on the association between tombstone cost and longevity.

---

## [Decision Letter · Decision Letter 3]

12 Oct 2023

PONE-D-23-07060R3Tombstone cost and longevity: the San Pedro Cemetery Museum in Medellín in ColombiaPLOS ONE

Dear Dr. O'Donovan,

Thank you for submitting your manuscript to PLOS ONE. After careful consideration, we feel that it has merit but does not fully meet PLOS ONE’s publication criteria as it currently stands. Therefore, we invite you to submit a revised version of the manuscript that addresses the points raised during the review process.

As you can read, Reviewer 1 suggest that you please address the concerns previously expressed in terms of the contextualization and interpretation of your results. I suggest that you address these comments in the discussion, including your reflections on the topics and how future research could fully address them.==============================

We look forward to receiving your revised manuscript.

Kind regards,

Juan Pablo Gutierrez

Academic Editor

PLOS ONE

Journal Requirements:

Reviewers' comments:

Reviewer's Responses to Questions

**Comments to the Author**

1. If the authors have adequately addressed your comments raised in a previous round of review and you feel that this manuscript is now acceptable for publication, you may indicate that here to bypass the “Comments to the Author” section, enter your conflict of interest statement in the “Confidential to Editor” section, and submit your "Accept" recommendation.

Reviewer #1: All comments have been addressed

Reviewer #2: All comments have been addressed

2. Is the manuscript technically sound, and do the data support the conclusions?

Reviewer #1: Partly

Reviewer #2: Yes

3. Has the statistical analysis been performed appropriately and rigorously? 

Reviewer #1: Yes

Reviewer #2: Yes

4. Have the authors made all data underlying the findings in their manuscript fully available?

Reviewer #1: Yes

Reviewer #2: Yes

5. Is the manuscript presented in an intelligible fashion and written in standard English?

Reviewer #1: Yes

Reviewer #2: Yes

6. Review Comments to the Author

Reviewer #1: The new version is very similar to the previous one. I suggest that authors not only analyze data but postulate explanations for what they observe. There are sociological and anthropological studies that could help explain the findings. This is important to guide future similar research. I hope they do not give up on theoretically deepening the results obtained.

Reviewer #2: (No Response)

7. PLOS authors have the option to publish the peer review history of their article (what does this mean?). If published, this will include your full peer review and any attached files.

Reviewer #1: No

Reviewer #2: No

---

## [Author Response · Author response to Decision Letter 3]

13 Oct 2023

Tombstone cost and longevity: the San Pedro Cemetery Museum in Medellín in Colombia

Response to reviewers

Reviewer comments

Reviewer #1

[Comment 1] The new version is very similar to the previous one. I suggest that authors not only analyze data but postulate explanations for what they observe. There are sociological and anthropological studies that could help explain the findings. This is important to guide future similar research. I hope they do not give up on theoretically deepening the results obtained. 

[Response 1] In the Discussion section in the revised manuscript, we have added a paragraph about Medellín that should help the reader place the results of the study in context. Please see below. We do not want to label or stigmatize the poor and the marginalized and we reiterate that we have no way of knowing whether the young subjects in the present study were victims of gang-related crime.

“Medellín changed from being a small town to a city in the 1970’s (Museo Casa de la Memoria, 2023). The population of Medellín doubled between 1965 and 1985 because of three waves of migrants: families fleeing political violence in the countryside; people looking to work and study and people looking for a better quality of life; and, people attracted by the government’s promotion of urban developments (Museo Casa de la Memoria, 2023). The authorities struggled to cope with the growing population and there was an industrial crisis, high unemployment, and a large black market controlled by drug-traffickers (Museo Casa de la Memoria, 2023). Medellín was characterized by two phenomena of urban growth in the 1970’s: there was planned and regulated urbanization along the west side of the River Medellín and there was informal and illegal development on the hillsides of the north of the city (Museo Casa de la Memoria, 2023). More than 57,000 young people were murdered in Medellín between 1979 and 2018 and people aged 18 to 24 years are still the main victims of murder (Alcaldía de Medellín, 2023). This long cycle of violence has led many academics to conclude that the poor and the marginalized are destined to join a gang and die young (Alcaldía de Medellín, 2019). Murder is the ultimate manifestation of taking the law into one’s own hands in poor and marginalized communities in Medellín and elsewhere in Latin America (Alcaldía de Medellín, 2019) and an investigation of 12 young murder victims found that 11 of them had experienced threats and violent situations weeks, months, and even years before their deaths (Casa de las Estrategias, 2017). It is the precarious nature of many young people’s lives in Medellín that leads to violence (Alcaldía de Medellín, 2019). Precarious economic conditions mean that many young people have limited access to essential goods and services (Alcaldía de Medellín, 2019). Precarious social conditions mean that many young people do not enjoy the same social fabric and the same social mobility as others (Alcaldía de Medellín, 2019). Precarious cultural conditions mean that many young people lack appropriate role models and are exposed to perverse norms and values (Alcaldía de Medellín, 2019). Precarious access to formal justice means that violence is often used to settle disputes (Alcaldía de Medellín, 2019). More than 200 young people in Medellín took part in workshops and semi-structured interviews between 2018 and 2019, including young people who had committed murder (Alcaldía de Medellín, 2019). Nine external factors were found to be associated with violent crime and murder: selling and consuming narcotic drugs; the need or desire to earn money; dropping out of school; displacement by violence; coming from a violent family or violent community; living in a community run by gangs; ‘invisible borders’ and limited mobility between communities; the opportunity to join a local gang; and, the exposure to uncontrolled violence (Alcaldía de Medellín, 2019). Four profiles of young people at risk of being the victim or the perpetrator of murder were also identified: members of criminal gangs; members of less formal groups of drug users and petty criminals; independent criminals who are not affiliated with a gang; and, young people outside of the criminal world who kill after taking drugs or getting into an argument (Alcaldía de Medellín, 2019). Gang-related crime was the main cause of murder in young people in Medellín in 2020, 2021, and 2022 (Alcaldía de Medellín, 2023), but we have no way of knowing whether the young subjects in the present study were victims of gang-related crime.”

References

Alcaldía de Medellín. (2019). Factores que inciden en el homicidio de jóvenes en Medellín: Propuesta de acción en clave de gobernanza colaborativa. Retrieved from https://www.medellin.gov.co/es/wp-content/uploads/2022/10/factores-que-inciden-en-el-homicidio-de-jovenes.pdf

Alcaldía de Medellín. (2023). Sistema de Información para la Seguridad y la Convivencia (SISC). Retrieved from https://www.medellin.gov.co/es/secretaria-seguridad/sisc/

Casa de las Estrategias. (2017). Los jóvenes destinados al homicidio en Medellín. In J. Giraldo (Ed.), Territorios y sociabilidades violentas. Santo Domingo, San Juan, Sao Paulo, Cali y Medellín (pp. 177-197). Medellín: Universidad EAFIT.

Museo Casa de la Memoria. (2023). Década de los 70. Retrieved from https://www.museocasadelamemoria.gov.co/medellin708090/decada-del-70/

---

## [Editor Report · Decision Letter 4]

19 Oct 2023

Tombstone cost and longevity: the San Pedro Cemetery Museum in Medellín in Colombia

PONE-D-23-07060R4

Dear Dr. O'Donovan,

We’re pleased to inform you that your manuscript has been judged scientifically suitable for publication and will be formally accepted for publication once it meets all outstanding technical requirements.

Kind regards,

Juan Pablo Gutierrez

Academic Editor

PLOS ONE
---

## [Editor Report · Acceptance letter]

8 Jan 2024

PONE-D-23-07060R4 

PLOS ONE

Dear Dr. O'Donovan, 

I'm pleased to inform you that your manuscript has been deemed suitable for publication in PLOS ONE. Congratulations! Your manuscript is now being handed over to our production team.

Kind regards, 

on behalf of

Dr. Juan Pablo Gutierrez 

Academic Editor

PLOS ONE